# Design of a Carangiform Swimming Robot through a Multiphysics Simulation Environment

**DOI:** 10.3390/biomimetics5040046

**Published:** 2020-09-30

**Authors:** Daniele Costa, Giacomo Palmieri, Matteo-Claudio Palpacelli, David Scaradozzi, Massimo Callegari

**Affiliations:** 1Department of Industrial Engineering and Mathematical Sciences, Polytechnic University of Marche, 60131 Ancona, Italy; d.costa@univpm.it (D.C.); g.palmieri@univpm.it (G.P.); m.c.palpacelli@univpm.it (M.-C.P.); 2Department of Information Engineering, Polytechnic University of Marche, 60131 Ancona, Italy; d.scaradozzi@univpm.it

**Keywords:** biomimetics, underwater robots, robotics, computational fluid dynamics, multibody systems, multiphysics simulations

## Abstract

Bio-inspired solutions devised for autonomous underwater robots are currently being investigated by researchers worldwide as a way to improve propulsion. Despite efforts to harness the substantial potential payoffs of marine animal locomotion, biological system performance still has far to go. In order to address this very ambitious objective, the authors of this study designed and manufactured a series of ostraciiform swimming robots over the past three years. However, the pursuit of the maximum propulsive efficiency by which to maximize robot autonomy while maintaining acceptable maneuverability ultimately drove us to improve our design and move from ostraciiform to carangiform locomotion. In order to comply with the tail motion required by the aforementioned swimmers, the authors designed a transmission system capable of converting the continuous rotation of a single motor in the travelling wave-shaped undulations of a multijoint serial mechanism. The propulsive performance of the resulting thruster (i.e., the caudal fin), which constitutes the mechanism end effector, was investigated by means of computational fluid dynamics techniques. Finally, in order to compute the resulting motion of the robot, numerical predictions were integrated into a multibody model that also accounted for the mass distribution inside the robotic swimmer and the hydrodynamic forces resulting from the relative motion between its body and the surrounding fluid. Dynamic analysis allowed the performance of the robotic propulsion to be computed while in the cruising condition.

## 1. Introduction

The comparative biomechanics of motion through water has attracted the attention of biologists and engineers for a long time, and recent decades have shown a significant growth in the study of aquatic animal locomotion. Attempts to design autonomous underwater vehicles (AUVs) capable of moving similarly to marine mammals and fish have been a result of the superior performance of biological swimmers, both in terms of efficiency and maneuverability [1]. Hence, several prototypes of bio-inspired robots have been manufactured worldwide, and extensive reviews are presented in [1,2]. The possibility of replicating the swimming modes that have evolved over thousands of years depends on an understanding of the fluid mechanic principles of aquatic animals’ locomotion. According to swimming mechanics, propulsive thrust originates from the momentum transfer due to the relative motion between a fish’s body and the surrounding water [3]. Particularly, body and caudal fin (BCF) swimmers generate the necessary thrust force by undulating their tails and caudal fins following specific patterns [4], such as Lighthill’s travelling wave [5]. The BCF classification is further expanded into five swimming modes characterized by the percentage of the body bent by the fish to propel itself, as shown in Figure 1.

The highest propulsive efficiency is achieved in thunniform locomotion, where the body undulation is confined to the rigid and flapping caudal fin. On the other hand, anguilliform swimmers experience the uppermost maneuverability as they bend their bodies according to different patterns called swimming gaits. Subcarangiform and carangiform locomotions are in the middle; here, the percentage of the body that participates in thrust generation shrinks from one-half to the last third, thus involving the tail of the fish [3]. Therefore, these swimming modes represent a compromise in terms of efficiency and maneuverability for an underwater robot propelled by a biomimetic thruster.

In the last few years, the authors of this paper have designed, manufactured, and tested a series of ostraciiform biomimetic vehicles [1,6], both for research and educational purposes, as shown in Figure 2. Here, the robot thruster consists of an oscillating plate shaped like a caudal fin and hinged to the rigid forebody. Ostraciiform locomotion is the least efficient among the BCF swimming modes [1]; however, since the number of moving parts is very limited, the resulting architecture is inexpensive and easy to fabricate and seal.

The trade-off between the conflicting demands of maneuverability and propulsive efficiency caused the authors to adjust and improve upon their design, moving from the ostraciiform to the carangiform swimming mode, as detailed in the present work. As stated in [1], the motion law generally adopted in modeling the tail undulation of carangiform swimmers is Lighthill’s travelling wave [5], a harmonic function with an amplitude that increases towards the caudal fin and that describes the shape of the tail as a function of time. Several robotic fish prototypes propelled by carangiform locomotion have been manufactured in the past few decades [1,7,8]. In addition to the novel soft robotics design method [9], the most common solution consists of a rigid head hinged to a piecewise flexible tail driven by multijoint serial mechanisms. Here, each revolute joint along the kinematic chain is directly actuated by a dedicated servomotor. Thus, in order to comply with the tail deformation predicted by Lighthill’s travelling wave, it is easy to show how each link of the mechanism must oscillate following a harmonic motion law, such as [10,11]:(1)$$p_{j}\left(t\right)=a_{j}\sin\left(2\pi ft+\varphi_{j}\right)$$
where *p_j_* is the absolute angle of link *j* with respect to a reference frame attached to the robot head; *a_j_* is the oscillation amplitude; *f* is its frequency, which is the same for all the links; and *φ_j_* is the constant phase shift between the harmonic motion laws of the links. Position control is then required to comply with the non-linear function (1) and to synchronize the servomotors in order to maintain the constant angular position difference between the links. Furthermore, each servomotor along the tail mechanism must be properly sealed, while watertight connections must be installed on both the joint shafts and on the electrical cables, leading to increases of the structure’s inertia, encumbrance, and possibility of failure.

On the basis of the aforementioned considerations, the main purpose of this paper is to design a transmission system for a carangiform swimming robot. The devised solution must be capable of driving the tail mechanism by means of a single rotary actuator while complying with Lighthill’s travelling wave. Aside from inertia and encumbrance reduction, the main improvement of the proposed solution will be the inherent synchronization of the tail links; in fact, both the phase differences between the motion laws (1), as well as their oscillation frequencies, will be maintained at a constant level by mechanical constraint of the transmission system. Moreover, waterproofing issues will be minimal, because the tail section will be fully flooded and only one motor will need to be sealed.

In order to design an actuation system that can predict robot swimming performance, the propulsive forces generated by the biomimetic thruster (i.e., the caudal fin) must be accurately quantified. For this purpose, computational fluid dynamics (CFD) analysis represents an invaluable tool in computing the hydrodynamic loads arising from fluid–structure interactions [12,13]. Furthermore, CFD predictions are generally expressed as a function of a few non-dimensional parameters that take into account both the geometry and the kinematics of a thruster. This feature allows a few long-lasting simulations to be performed, then applies the numerical solutions to many different scenarios. However, in order to assess the dynamic performance of the vehicle, the propulsive forces obtained by CFD analysis must be integrated in a multibody model that accounts for both mass distribution and hydrodynamic effects (e.g., added mass and viscous damping). As a matter of fact, multibody techniques can be easily exploited to study large classes of biomimetic underwater vehicles, so long as they have a rigid forebody and a multijoint robotic tail. In other words, CFD and multibody analysis represent the backbone of a multiphysics simulation environment that can be successfully exploited in order to design and optimize both the propulsive system and mass distribution of a given robot [14].

## 2. Materials and Methods

In carangiform locomotion, the wave-shaped undulations are mostly restricted to the last third posterior part of the body and increase sharply in the caudal fin area [3]. This swimming mode is typical of many aquatic animals, such as the Atlantic mackerel (or *Scomber Scombrus*) [15]. Fish in the Scombridae family are characterized by a streamlined body with a homocercal caudal fin and can achieve high swimming speeds. The geometric and inertial features of the robotic fish designed in this paper are modeled after a mackerel, as shown in Figure 3. The final assembly is a 2:1 scale of the real fish—the robot is 800 mm long and its tail spans over the last 300 mm.

### 2.1. Tail Kinematics and Optimization Criteria

The transmission system proposed in this paper drives a multijoint tail mechanism designed in order to comply with Lighthill’s travelling wave function:(2)$$y_{TW}\left(t\right)=\left(c_{1}x+c_{2}x^{2}\right)\sin\left(kx+2\pi ft\right)$$
where *y_TW_* is the lateral displacement; *x* is the distance from the connection point (Figure 4a); *k* and *f* are the wave number and frequency, respectively; and *c*_1_ and *c*_2_ are kinematic parameters called linear and quadratic amplitude envelopes, respectively. In order to approximate the continuous bending deformation expressed by Equation (2) by means of a piecewise flexible mechanism, the authors propose a discretization method widely used in former prototypes [1,10]: the tail motion function (2) is discretized in *M* postures *h_T_*(*x*,*i*), (*i* = 0 … *M*−1) over time, as shown in Figure 4b; then, each posture is approximated by a pose of the multijoint mechanism. The number of links of the mechanism is limited to 3 in this work, because the links must be long enough to house the components of the transmission system, and a minimum length of 100 mm is imposed on each link for an overall length of 300 mm. The first link is hinged to the fish head at the connection point (Figure 4a), while the third link is the caudal fin. Figure 5 shows the approximation of a tail posture, where (*x_k_,y_k_*) (*k* = 0 … 2) are the coordinates of joint *k*, *(x_3_,y_3_)* is the endpoint of the third link, and *p_ij_* (*i* = 1 … *M*−1, *j* = 1 … 3) are the absolute angles of link *j* relative to the head in the *i*th tail posture. Hence, the pose of the mechanism can be represented by the piecewise linear function *g*(*x*,*i*), defined as:(3)$$g\left(x,i\right)=\{\begin{array}{c} \begin{array}{cc} y_{0}+p_{i,1}\left(x-x_{0}\right) & x_{0}\leq x<x_{1} \end{array}\\ \begin{array}{cc} y_{1}+p_{i,2}\left(x-x_{1}\right) & x_{1}\leq x<x_{2} \end{array}\\ \begin{array}{cc} y_{2}+p_{i,3}\left(x-x_{2}\right) & x_{2}\leq x<x_{3} \end{array} \end{array}$$

In order to compute the pose of the mechanism that best approximates the tail postures *h_T_*(*x*,*i*), a root mean square error criterion is usually adopted [1]. However, since the main purpose is to approximate Equation (2) while staying true to the real fish dynamics, the added mass method in [10] is preferred and implemented to solve the optimal pose of the mechanism and to compute the unknown angles *p_ij_*. In carangiform locomotion thrust is generated, since the propulsive wave travels along the body of the fish—momentum is transferred by the tail motion to the surrounding water, which in turn develops a reaction force pushing the swimmer. The mass of the water passing backwards is called added mass and the magnitude of the thrust force depends on it. Therefore, when using a multijoint serial mechanism, the optimal configuration is computed by minimizing the difference between the added mass pushed by the linkage and by the real tail, which leads to the following expression [10]:(4)$$e_{i}\left(p_{i1},p_{i2},p_{i3}\right)=\int_{x_{0}}^{x_{3}}\left|h_{T}\left(x,i\right)-g\left(x,i\right)\right|dx$$
where *e_i_* is the error function, which depends on the absolute angle *p_ij_* in the *i*th posture. Expression (4) can then be minimized as a function of *p_ij_* for each value of *i*, thus computing the optimal pose of the tail mechanism, as shown in Figure 6. As stated before, each link of the kinematic chain oscillates following a harmonic function of time. Thus, the optimal values of the angles *p_i_*_1_, *p_i_*_2_, and *p_i_*_3_ (*i* = 0 … *M*−1) can be interpolated to closely resemble the sinusoidal functions *p_j_*(*t*) (*j* = 1 … 3) stated by (1), as shown in Section 3.

### 2.2. Elements of the Cam-Joint-Based Transmission System

In [6], the authors of this paper designed a transmission mechanism to drive the caudal fin of an ostraciiform swimming robot. The core of the system is the spatial cam kinematic joint shown in Figure 7a. The aim is to transform the input rotary motion of the motor into a harmonic oscillation of the output shaft. The driving disk A has a drive sphere B. which engages in a rectangular groove of the driven member C. The oscillating output shaft is rigidly connected with the element C and is also able to pivot freely in the support block D. The arrangement is compact and the axis of rotation of the output is perpendicular to the input. The maximum rotation of the output shaft is twice the angle *θ*_0_ shown in Figure 7b.

More generally, the output angle *θ* and the motor rotation *φ* are related by the following expression:(5)$$\tan\theta =\frac{h}{L}\cos\varphi =\lambda \cos\varphi$$
where *h* and *L* are the geometric parameters outlined in the kinematic scheme of Figure 7b. It is easy to prove that Expression (5) can be approximated by a pure sinusoidal function of time if the motor spins at a constant speed *ω* and if *λ* is sufficiently smaller than one:(6)$$\begin{array}{ccc} \theta \approx \lambda \cos\varphi =\lambda \cos\left(\omega t\right) & \dot{\varphi }=\omega & \lambda \approx \theta_{0} \end{array}$$

With the proposed transmission system, the effort required for the motor control system is reduced, since a constant velocity setting for the motor is able to generate a harmonic oscillation of the output shaft. Otherwise, a direct drive of the output shaft would require position control to comply with a non-linear function of time. The mechanical system presented in this section is also the core of the transmission mechanism designed by the authors to drive the articulated robotic tail presented in this paper. 

### 2.3. Computational Fluid Dynamics Analysis

In order to design the vehicle actuation system and to predict its behavior, it is necessary to compute the propulsive forces and torque generated by the caudal fin as the robotic tail undulates according to the travelling wave-approximated pattern. To this end, CFD techniques are used to solve the velocity and pressure fields around the caudal fin once its motion law is derived following the optimization method in Section 2.1. As a matter of fact, the caudal fin coincides with the third link of the robotic tail, performing a harmonic roto-translation characterized by a constant phase shift between the pitching and heaving components of motion, as shown in Section 3.

In this work, the CFD analysis is performed using an in-house-developed research code named MIGALE, based on the discontinuous Galerkin (DG) space discretization [16,17], to solve the Reynolds-averaged Navier–Stokes equations (RANS). DG variational methods combine features of the finite elements, such as the element-wise polynomial representation of the solution, and of the finite volumes as well, such as the computation of the numerical fluxes at the mesh element interfaces. The advantage of using such methods is related to the compact stencil of the space discretization, which is independent from the order of the employed polynomial approximation, as the solution does not need to be continuous at mesh element interfaces. This fact makes DG methods well-suited for the implementation of high-order implicit time integration schemes. Moreover, DG can provide very accurate solutions on curved and hybrid computational grids.

Here, the authors have employed the two-dimensional and incompressible version of the DG code, which is suitably extended to deal with a moving reference frame [17] to account for the fin oscillation. In this way, the computational complexity of the solving algorithm is reduced when compared to the dynamic mesh boundary condition commonly used in commercial CFD codes. The two-dimensional mesh was created using 4670 elements and is suitably curved to represent the leading edge. Figure 8a shows that the mesh is finer near the solid surface, as well as along the wake. The RANS equations were closed using a Spalart–Allmaras turbulence model [18] and the governing equations were discretized in time using a third-order, three-stages, linearly implicit Runge–Kutta scheme. Fifth-order polynomials were used to represent the solution within each element, resulting in sixth-order space discretization, while the time step size was set to one-thousandth of the caudal fin oscillation period to ensure the results are independent of the time discretization. Finally, a multicore (32 processors) parallel computer system was employed to compute the numerical solution, taking about 32–48 h to complete each simulation.

The numerical analysis is performed on two different foils, which are obtained by slicing the caudal fin with two horizontal planes at 25% and 75% of its span *B*/2, as shown in Figure 8b. The boundary conditions are set in order to match the foil kinematics; particularly, the origin of the moving reference frame coincides with the revolute joint connecting the second link of the tail to the caudal fin, as shown in Figure 8b. In other words, the fin roto-translating motion is generated by the moving reference frame, while the fluid free stream velocity boundary condition accounts for the swimming motion of the robot. The results of the analysis and the propulsive forces and torque generated by the three-dimensional thruster are detailed in Section 3.

### 2.4. Multibody Model

Figure 9a shows a cylindrical body moving in the surrounding water; the reference frame *Σ_b_*, *O_b_* − *x_b_y_b_z_b_* is attached to the body. The velocity of the origin *O_b_* is expressed in the body frame *Σ_b_* by the vector ***ν*_1_** = [*u v w*]^T^; likewise, vector ***ν*_2_** = [*p q r*]^T^ represents its angular velocity [19]. In this paper, the authors have focused their analysis on plane motion. In this case and according to the Newton–Euler formulation, the dynamics equations can be written as:(7)$$\begin{array}{c} m\left(\dot{u}-vr\right)=X\\ m\left(\dot{v}+ur\right)=Y\\ I_{z}\dot{r}=N \end{array}$$
where *m* is the body mass and *I_z_* is the *z* principal moment of inertia, which is computed under the hypothesis that the frame *Σ_b_* is coincident with the body central frame. The right side of Equation (7) accounts for the hydrodynamic loads applied to a rigid body moving in the surrounding fluid. A rigorous analysis of an incompressible flow would require the solution of the Navier–Stokes equations; however, if the velocities are reasonably low, most of the hydrodynamic effects have no significant influence on the resulting motion. Moreover, if the body features three planes of symmetry, the terms in the right side of Equation (7) can be linearized [19,20], leading to the following expression:(8)$$\begin{array}{c} X=-X_{\dot{u}}\dot{u}-Y_{\dot{v}}vr-X_{u\left|u\right|}u\left|u\right|\\ Y=-Y_{\dot{v}}\dot{v}+X_{\dot{u}}ur-Y_{v\left|v\right|}v\left|v\right|\\ N=-N_{\dot{r}}\dot{r}-N_{r\left|r\right|}r\left|r\right| \end{array}$$
where the subscripts with the derivatives of the velocity identify the coefficients of the added mass hydrodynamic loads, while the velocity subscripts appear in the damping coefficients. Table 1 collects the expressions of the terms in Equation (8) for a cylinder with a radius *R*, length *L*, and mass *m*.

Figure 9b shows the multibody model of a swimming robot, composed of a rigid head hinged to a three-joint tail linkage ending with a caudal fin. Body reference frames *Σ_b,i_* (*i* = 0 … 3) are attached to the rigid bodies of the model; the zero index refers to the robot head, while non-zero indexes identify the tail links. Each body of the assembly, except the caudal fin, is approximated by a cylinder and is subjected to hydrodynamic forces coming from Equation (8). The propulsive forces and torque *F_T_*, *F_L_*, and *M*, are applied to the fin. Following the convention widely adopted in other works [3], the thrust force component is aligned with the swimming direction, coincident with the free stream velocity vector in the CFD analysis.

Dynamics Equations (7) and (8) are solved by using Adams/View by MSC Software^®^ (Newport Beach, CA, USA). The complete modeling procedure and the computed results are shown in Section 3.3.

## 3. Results

By following the optimization method presented in Section 2.1, each tail posture can be approximated by a pose of the linkage. The computed values of the absolute angles *p_i_*_1_, *p_i_*_2_, *p_i_*_3_ (*i* = 0 … *M*−1), which describe the orientation of the links in *M* postures, can then be interpolated, leading to the following expressions:(9)$$\begin{array}{c} p_{1}\left(t\right)=A_{1}\;\sin\left(2\pi ft+\Delta_{1}\right)\\ p_{2}\left(t\right)=A_{2}\;\sin\left(2\pi ft+\Delta_{2}\right)\\ p_{3}\left(t\right)=A_{3}\;\sin\left(2\pi ft+\Delta_{3}\right) \end{array}$$
where the respective values of the amplitudes *A_j_* and phase shifts Δ*_j_* are presented in Table 2. The curves showing the results of the interpolation are gathered in Appendix B.

### 3.1. Functional Design of the Transmission System

The robotic tail presented in this paper has the same kinematics of a 3R planar mechanism, namely three links connected by as many revolute joints driven by a single rotary motor. In order to drive its links while complying with the harmonic functions (1), a transmission system is integrated into the tail assembly. The core of the devised solution is the spatial cam kinematic joint discussed in Section 2.2. As a matter of fact, each revolute pair in the linkage is driven by a cam joint, which is suitably designed to transform the continuous rotation of the input shaft into the *j*th harmonic law (1). As stated before, the aim of the authors is to drive the whole robotic tail by means of a single motor installed in the robot frame, i.e., its rigid head. As a consequence, the three cam joints are driven by a single shaft that passes through the whole robotic tail while spinning at constant angular velocity. In this way, all the links oscillate with the same frequency *f*, fulfilling the synchronization requirement between the revolute pairs of the linkage. However, in order to allow the driving shaft of the cam joints to follow the robotic tail as it bends according to the approximated wave pattern, a homokinetic joint is required at each joint to transfer the driving torque, while the preserving angular freedom and constant rotational velocity of the shaft, as shown in Figure 10. In this paper, double Cardan joints are selected and used in homokinetic configuration [21]. As an assembly, homokinetic double Cardan joints require a centering element that maintains equal angles between the driven and driving shafts, as also stated by the “homokinetic plane” scheme. According to the theory of shaft coupling designed to produce a constant-velocity transmission, the centering element must share the same plane of symmetry with the one between the input and output shafts, generally called the “bisecting plane” or “homokinetic plane”; this plane must also contain the intersection of the axis of the joints forming the double Cardan [22]. Figure 11a shows the solution used in this paper. The plane of symmetry of each double Cardan joint coincides with the homokinetic plane, which the axis of the corresponding revolute joint of the robotic tail lies on. Furthermore, the centering element is composed of two parts that may slide relative to each other by means of a prismatic joint. In this way, the length of the centering element is passively adjusted as the linkage changes its configuration according to the tail undulation, while maintaining its mid-section coincident with the symmetry plane of the coupling, fulfilling the mentioned requirement of constant rotational velocity. 

In order to size the spatial cam joints and generate the harmonic laws (9), the following equations must be solved:(10)$$\begin{array}{c} p_{1}\left(t\right)=\theta_{1}\approx \lambda_{1}\cos\left(\omega t+\delta_{1}\right)\\ p_{2}\left(t\right)=\theta_{1}+\theta_{2}\approx \lambda_{1}\cos\left(\omega t+\delta_{1}\right)+\lambda_{2}\cos\left(\omega t+\delta_{2}\right)\\ p_{3}\left(t\right)=\theta_{1}+\theta_{2}+\theta_{3}\approx \lambda_{1}\cos\left(\omega t+\delta_{1}\right)+\lambda_{2}\cos\left(\omega t+\delta_{2}\right)+\lambda_{3}\cos\left(\omega t+\delta_{3}\right) \end{array}$$
where *θ*_1_, *θ*_2_, and *θ*_3_ are the output angles of the cam joint mechanisms shown in Figure 10; *λ_j_* and *δ_j_* are their functional parameters and initial rotations, respectively; whereas *ω* is the motor velocity. The approximated expression of the output angles (6) is used. By solving the first row of (10), this immediately results in:(11)$$\begin{array}{ccc} \lambda_{1}\approx A_{1} & \delta_{1}=\Delta_{1}=0 & \omega =2\pi f \end{array}$$

In order to compute the values of *λ*_2_ and *δ*_2_, from Equation (10) it can be found that:(12)$$A_{2}\cos\left(\omega t\right)\sin\Delta_{2}+A_{2}\sin\left(\omega t\right)\cos\Delta_{2}\approx \left(\lambda_{1}+\lambda_{2}\cos\delta_{2}\right)\cos\left(\omega t\right)-\lambda_{2}\sin\left(\omega t\right)\sin\delta_{2}$$
which leads to:(13)$$\begin{array}{c} \lambda_{2}\cos\delta_{2}\approx A_{2}\sin\Delta_{2}-\lambda_{1}\\ \lambda_{2}\sin\delta_{2}\approx -A_{2}\cos\Delta_{2} \end{array}$$

Equation (13) can then be solved as a function of the known parameters *A*_2_, Δ_2_, and *λ*_1_, leading to:(14)$$\begin{array}{c} \tan\delta_{2}\approx \frac{A_{2}\cos\Delta_{2}}{\lambda_{1}-A_{2}\sin\Delta_{2}}\\ \lambda_{2}{}^{2}\approx A_{2}{}^{2}+\lambda_{1}{}^{2}-2A_{2}\lambda_{1}\sin\Delta_{2} \end{array}$$

Such result allows us to size the second cam joint and to set its initial rotation with respect to the former. The same procedure allows us to solve the third row in (10), and thus compute the joint parameters *λ*_3_ and *δ*_3_ as functions of the known quantities *λ*_1_, *λ*_2_, *δ*_2_, *A*_3_, and Δ_3_:(15)$$\begin{array}{c} \lambda_{3}\cos\delta_{3}\approx A_{3}\sin\Delta_{3}-\lambda_{2}\cos\delta_{2}-\lambda_{1}\\ \lambda_{3}\sin\delta_{3}\approx -A_{3}\cos\Delta_{3}-\lambda_{2}\sin\delta_{2} \end{array}$$

Finally:(16)$$\begin{array}{c} \tan\delta_{3}\approx \frac{A_{3}\cos\Delta_{3}+\lambda_{2}\sin\delta_{2}}{\lambda_{1}+\lambda_{2}\cos\delta_{2}-A_{3}\sin\Delta_{3}}\\ \lambda_{3}{}^{2}\approx A_{3}{}^{2}+\lambda_{2}{}^{2}+\lambda_{1}{}^{2}-2A_{3}\lambda_{2}\sin\left(\Delta_{3}-\delta_{2}\right)-2A_{3}\lambda_{1}\sin\Delta_{3}+2\lambda_{1}\lambda_{2}\cos\delta_{2} \end{array}$$

Once the functional parameters *λ_j_* are computed, the geometric parameters of each cam joint, *h_j_* and *L_j_*, can be chosen according to Expression (5), which states that *λ_j_* = *h_j_/L_j_*; moreover, the radius of the drive spheres shown in Figure 7a,b must be large enough to house the transmission input shaft and prevent interferences with the rectangular grooves of the joint output member. However, as the sphere radius grows, the encumbrance of the cam joints rises as well. In order to overcome this constraint, the drive spheres are drawn with a cut shape, leaving only the small surface astride the contact line with the driven member in an oscillation cycle, whereas the remaining material, i.e., the part not involved in torque transmission, is ideally removed. As a matter of fact, contact occurs on the line resulting from the intersection of the drive sphere surface and plane *Π*, as shown in Figure 11b, where angle *θ*_0*j*_ is equal to the output oscillation amplitude [23]. Table 3 summarizes the geometric parameters of the cam joints designed in this paper.

Figure 12 compares the trends of the absolute angles *p_j_*(*t*), expressed by (9), with their approximations obtained by using the exact expressions (5) of the cam joints output angles *θ_j_* in Equation (10). The curves show a good agreement between the target trends and those achieved with the transmission system proposed here.

### 3.2. Propulsive Performance of the Caudal Fin

As the robotic tail undulates according to the approximated travelling wave function (9), the fin performs a harmonic roto-translation characterized by a constant phase shift *Λ* between the pitching and heaving components of motion. The fin rotation is already available in (9) as *p*_3_(*t*), whereas the translation motion law can be calculated as the lateral displacement *s* of the revolute joint that connects the fin to the former linkage member:(17)$$y_{2}\left(t\right)=d\sin\left(\theta_{1}\right)+d\sin\left(\theta_{1}+\theta_{2}\right)=Y_{0}\sin\left(2\pi ft+\Delta_{y}\right)$$
where the link length *d* is 100 mm, as discussed in Section 2.1, the translation amplitude *Y*_0_ measures 57 mm and Δ*_y_* is equal to −0.43. This follows a constant phase shift *Λ* of about 51 degrees.

The propulsive performance of a roto-translating foil depends on five kinematic parameters: the rotation and translation amplitudes, the frequency and phase shift between the respective harmonic motion laws, plus the position of the rotation axis. Therefore, a large campaign of numeric analyses is necessary to investigate the relative influence of the aforementioned quantities on the foil propulsive performance. Although the simulation results are presented in Appendix C, a sensitivity analysis such as the one just outlined is beyond the scope of this paper. According to the kinematics proposed in Section 3.1, four out of five parameters are known and constant. Therefore, only the foil oscillation frequency is left out.

Nevertheless, the first set of fluid dynamics analyses deals with the performance sensitivity of the Reynolds number, *Re*, a dimensionless parameter that accounts for the ratio between inertial and viscous forces. In this paper, *Re* ranges between 10^4^ and 10^6^, while the remaining kinematic parameters remain constant. The simulations show that both the efficiency and the thrust coefficients are poorly related to the Reynolds number, as reported in Table 4. The simulations confirm that beyond the critical *Re* for a roto-translating foil, the flow becomes turbulent and propulsive forces stabilize close to a steady-state value. Still, a slightly growing trend is observed for both the efficiency and the thrust coefficient as the Reynolds number increases. In fact, the higher the *Re*, the stronger the turbulent boundary layer surrounding the foil, thus preventing flow separations, even at the highest incidence angles. The low dependency on the Reynolds number allows the authors to focus their investigation on the other kinematic parameters. Indeed, the robotic fish designed in this paper should swim at *Re* values varying between 10^5^ and 10^6^, a range that is dependent on its speed, which in turns derives from the tail undulation frequency, as shown in Section 3.3.

It is convenient to express the influence of the oscillation frequency in dimensionless form by introducing the Strouhal number *St*, a fluid dynamics parameter widely used to characterize periodic flow phenomena. For a roto-translating foil, *St* is defined as:(18)$$St=f\frac{A}{U}\approx f\frac{2Y_{0}}{U}$$
where *U* is the free stream velocity and *A* is the wake width, which is approximated by twice the translation amplitude *Y*_0_ [24]. Figure 13 shows the propulsive efficiency and the average thrust coefficient as a function of the Strouhal number. The four lines refer to two different foils obtained by slicing the fin with two horizontal planes at 25% and 75% of its span, respectively, as explained in Section 2.2 and Figure 8. Despite the tapered shape of the fin, since both foils share the same translation amplitude due to the heaving rigid motion component, they are also characterized by the same values of *St*.

Figure 13 shows that the average thrust coefficient depends on the Strouhal number squared, which is true also for the lateral force, i.e., the force normal to the freestream velocity, as well as the torque coefficients. Therefore, the propulsive load coefficients can be expressed by the following function of time:(19)$$C_{pl}\approx K_{pl}St^{2}\sin\left(2\pi ft+\varphi_{pl}\right)$$
where *C_pl_* is a propulsive load coefficient, *K_pl_* is a proportionality factor, and *φ_pl_* is a phase constant. As a matter of fact, the CFD simulations proved that the trends in an oscillation cycle of both foil load coefficients can be approximated by harmonic functions, whose amplitudes depend on the square of *St*, whereas their phase shifts are independent from the aforementioned parameter. Since both foils share the same Strouhal number, the propulsive forces and torque resulting from the three-dimensional fin can be finally approximated by the following expression, which is widely used in aircraft wing design:(20)FT=12ρU2(CT,25%c25%+CT,75%c75%)B2FL=12ρU2(CL,25%c25%+CL,75%c75%)B2M=12ρU2(CM,25%c25%2+CM,75%c75%2)B2
where *F_T_*, *F_L_*, and *M* are the propulsive forces and torque already introduced in the multibody model in Section 2.4; *ρ* is the water density; whereas *c*_25%_ and *c*_75%_ are the foil chord lengths. Table 5 shows the values of the geometric quantities used in (20).

### 3.3. Multibody Analysis

In this paper, the dynamic analysis is performed by using MSC Adams/View software. The robot CAD model is imported in the multibody environment, as show in Figure 14, while the transmission mechanism designed in Section 3.1 is integrated into the tail of the robotic fish. Each link is covered by a rigid shell derived from the model skin surface. Since the aim of this work is to size the robot propulsive system in the cruising condition, a planar joint is applied to the center of mass of the fish in order to constrain its motion in the horizontal plane. Next, the hydrodynamic loads (8) resulting from the fluid–structure interaction are applied to the robot’s head and tail links, as detailed in Section 2.4. Finally, the propulsive forces and torque expressed by (20) are applied to the caudal fin using run time functions to compute their modulus through the simulations. Particularly, by substituting the Strouhal number formula (18) in System (20), the following expressions can be derived:(21)$$\begin{array}{c} F_{T}=\rho Y_{0}{}^{2}f^{2}B\left(K_{T,25\%}c_{25\%}+K_{T,75\%}c_{75\%}\right)\sin\left(2\pi ft+\varphi_{T}\right)\\ F_{L}=\rho Y_{0}{}^{2}f^{2}B\left(K_{L,25\%}c_{25\%}+K_{L,75\%}c_{75\%}\right)\sin\left(2\pi ft+\varphi_{L}\right)\\ M=\rho Y_{0}{}^{2}f^{2}B\left(K_{M,25\%}c_{25\%}{}^{2}+K_{M,75\%}c_{75\%}{}^{2}\right)\sin\left(2\pi ft+\varphi_{T}\right) \end{array}$$

The dynamic equations can then be solved to compute the unknown velocities *u*, *v*, and *r* as functions of a single parameter—the tail undulating frequency, which coincides with the motor angular velocity (except for the *2π* factor) as a consequence of the spatial cam joints used in the transmission system. The robot mass and inertia properties are detailed in Table 5.

The black line in Figure 15 indicates the average cruising velocity, which is reached after the acceleration transient, as a function of the motor revolute frequency. The corresponding maximum values of the driving torque are shown by the grey line. The swimming speed is expressed in relative form, meaning as a multiple of the total length of the fish (800 mm) per second, BL/s. The Strouhal number is almost constant at 0.235–0.237, close to the efficiency peaks predicted by the CFD analysis and shown in Figure 13. Those values also fit into the optimal range observed in swimming animals [25].

Video animations of the undulating tail mechanism and the swimming robotic fish are offered as Appendix A.

## 4. Discussion

Several approximations are introduced throughout the modeling and design process of the carangiform swimming robot presented in this work. Above all else, the authors neglected the contribution of tail undulation to the generation of thrust, meaning that the propulsive force is produced entirely by the roto-translation of the caudal fin. Unfortunately, the CFD code used in this work is unable to solve the velocity and pressure fields around multiple moving bodies, which would require the implementation of dynamic mesh boundary conditions and complex solving algorithms. Similar considerations stand for the characterization of the three-dimensional fin, which is derived by the propulsive performances of the component foils. As a matter of fact, the complete solution of the flow around a finite-span oscillating wing is beyond the scope of the present work, which is entirely focused on the design of propulsive system. Moreover, three-dimensional CFD simulations would require computational power and time one order of magnitude higher with respect to the variational methods employed by the authors.

Regarding the multibody analysis, it is worth noting that the average cruising speed has a linear dependency on the motor rotation frequency, as shown in Figure 15. As a matter of fact, a steady-state condition is reached when the damping force in the swimming direction (8) is balanced by the thrust generated by the caudal fin. While the damping force depends on the swimming speed squared, the thrust amplitude (21) contains the square of the motor frequency, resulting in the linear dependency in a steady-state condition. This last conclusion also explains the invariance of the Strouhal number with respect to the motor frequency. *St* is proportional to the ratio between *f* and the velocity modulus U, which almost coincides with the *u*-component because the lateral is significantly smaller. Furthermore, as for the Strouhal number value for the steady-state solution, the model produced by the authors fully reproduces the optimal swimming condition widely observed in marine animal locomotion. In other words, the solution predicted by the multibody analysis corresponds to the most efficient operating condition. Still, the efficiency peak is lower with respect to other BCF swimming modes, such as thunniform [1]. This is due to the design evolved by thunniform swimmers in terms of body shape and mass distribution, which is optimized for high-speed, efficient swimming, while is not well-suited to other actions, such as turning maneuvers and rapid acceleration. On the contrary, carangiform swimmers possess slender and flexible bodies, which allows them to perform turnabout maneuvers characterized by a normalized radius of curvature lower than one body length [1].

Finally, regarding the propulsive system, the transmission mechanism presented in this work is able to convert the continuous rotation at a constant frequency of a single motor in the travelling wave undulation of the carangiform robot tail. The adoption of the double Cardan joints allows the transmission shaft to spin with the same speed through the whole mechanism, which in turn results in the required synchronization between the oscillating members of the linkage. When compared to a robotic tail driven by multiple servomotors, the proposed solution reduces the effort of the control system, inertia, and encumbrance, while waterproofing issues are minimal, since only a single motor must be sealed.

The current project is now at the beginning of the prototyping phase. Future research will focus on the preparation of a CAD model of the major components, including the double Cardan joints, which will be printed by means of high-resolution stereolithography. Functional and sealing tests will follow as soon as the robotic tail is assembled. Moreover, future works will also be focused on the design and simulation of a transmission system capable of driving the pectoral fins of the proposed prototype. The devised solution will allow the robot to change its course according to its mission. At the same time, the multibody model will undergo extensive improvements in order to simulate both attitude variations and depth-changing maneuvers as well.

## Figures and Tables

**Figure 1 biomimetics-05-00046-f001:**
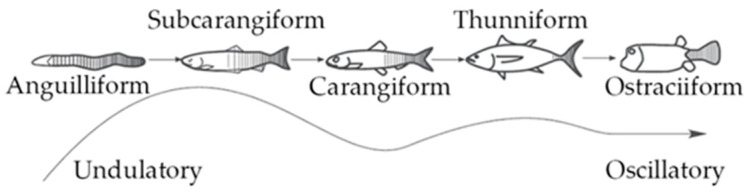
Body and caudal fin (BCF) swimming mode classifications [3].

**Figure 2 biomimetics-05-00046-f002:**
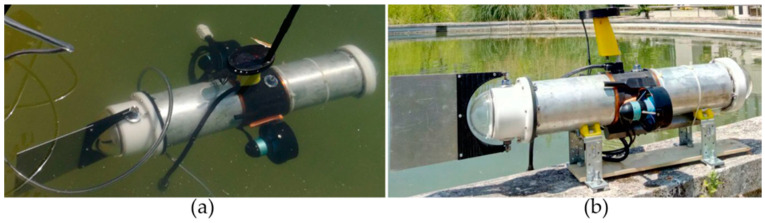
The ostraciiform swimming robot called “DORI”: (**a**) upper and (**b**) side views [1,6].

**Figure 3 biomimetics-05-00046-f003:**
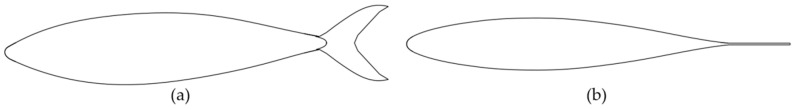
Sketches of the Atlantic mackerel modeled in this work: (**a**) side and (**b**) upper views [15].

**Figure 4 biomimetics-05-00046-f004:**
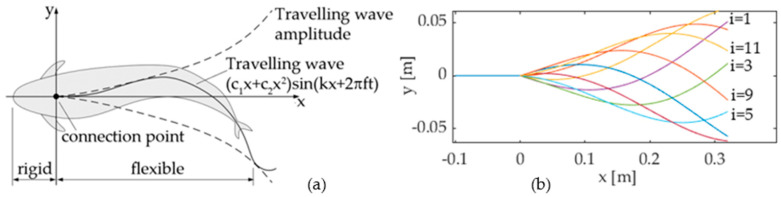
(**a**) Travelling wave pattern [1]. (**b**) Tail posture discretization (*c*1 = 0.2, *c*2 = 0; *k* = 7.5, *f* = 1, *M* = 18).

**Figure 5 biomimetics-05-00046-f005:**
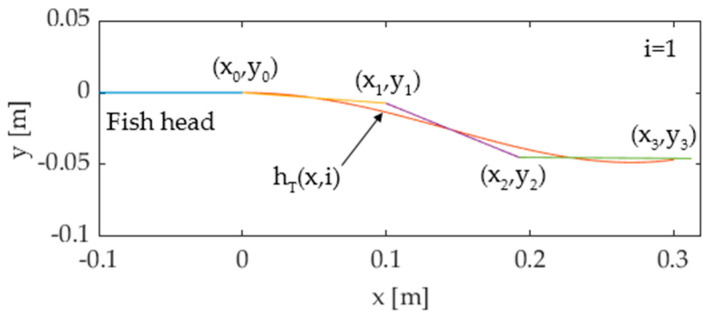
Tail posture approximated by the pose of a three-joint serial mechanism.

**Figure 6 biomimetics-05-00046-f006:**
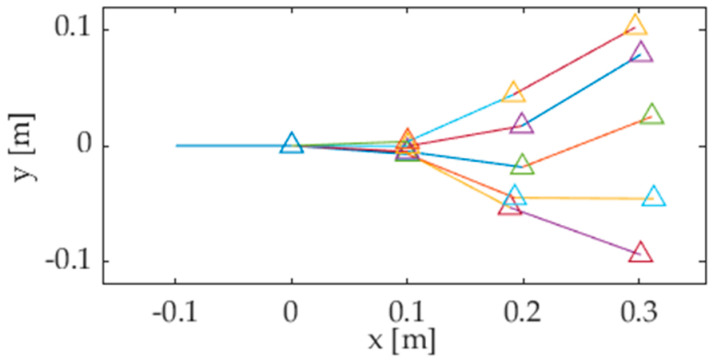
Optimal pose of the linkage corresponding to five tail postures.

**Figure 7 biomimetics-05-00046-f007:**
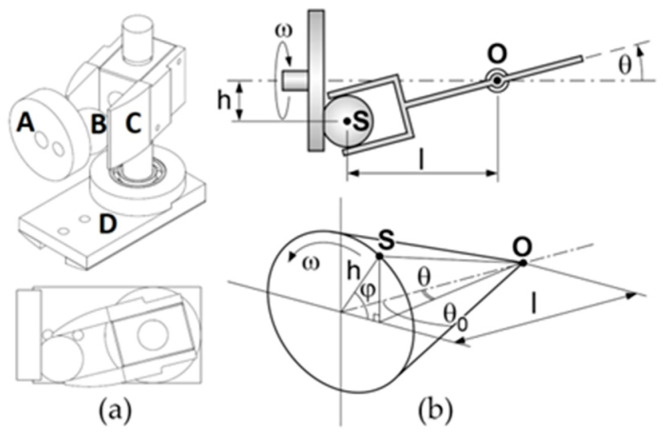
(**a**) CAD model of the spatial cam kinematic joint. (**b**) Functional scheme [1,6].

**Figure 8 biomimetics-05-00046-f008:**
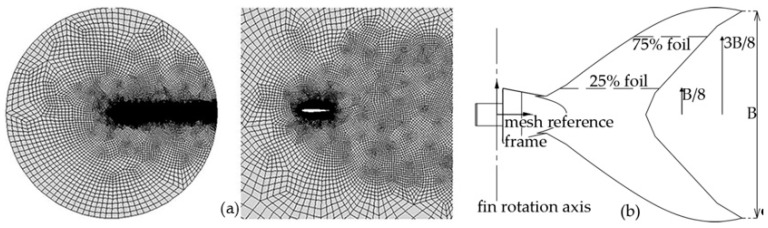
(**a**) Two-dimensional mesh used in computational fluid dynamics (CFD) analysis. (**b**) Fin geometry and foil definition.

**Figure 9 biomimetics-05-00046-f009:**
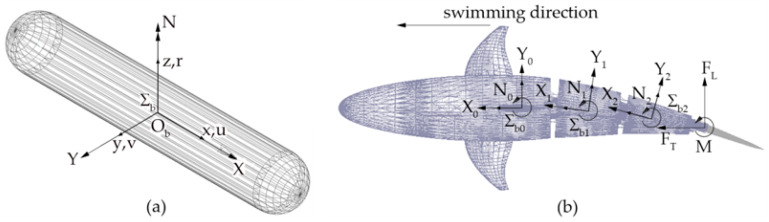
(**a**) Cylindrical body: hydrodynamic loads and body-frame convention [19]. (**b**) Robotic fish: hydrodynamic loads and propulsive forces.

**Figure 10 biomimetics-05-00046-f010:**
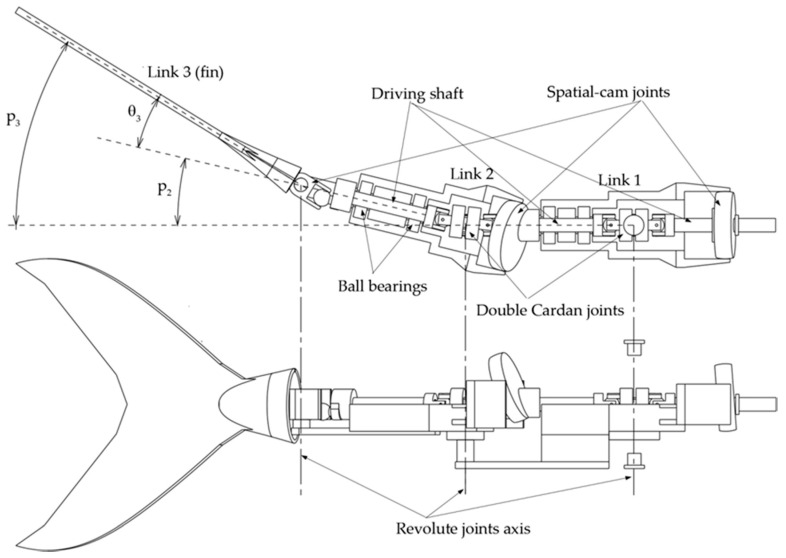
CAD model of the transmission system designed to drive the robotic tail at *t* = 0.

**Figure 11 biomimetics-05-00046-f011:**
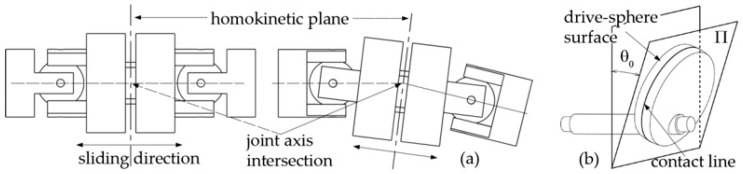
(**a**) Double Cardan joint: unbent tail (left), bent tail (right). (**b**) Spatial cam joint geometry.

**Figure 12 biomimetics-05-00046-f012:**
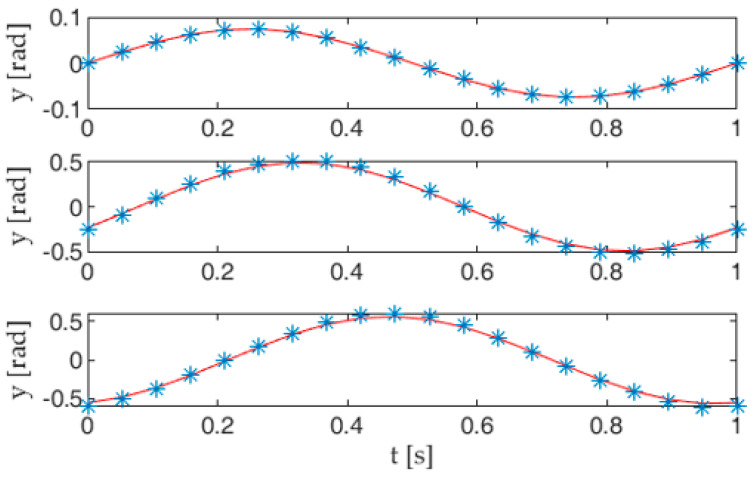
Trends of the absolute angles *p_j_* (solid lines) stated by Equation (9) compared to those obtained with the cam joint angles (dotted lines), as shown in (10).

**Figure 13 biomimetics-05-00046-f013:**
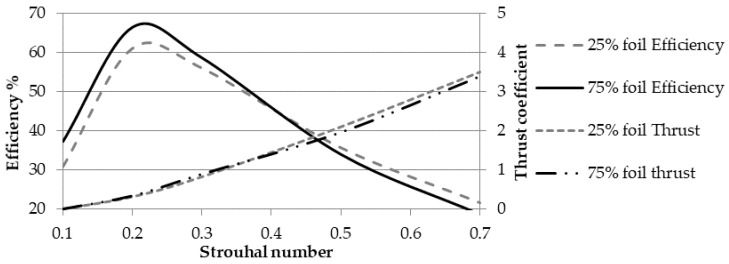
Propulsive efficiency and average thrust coefficient obtained by CFD analysis.

**Figure 14 biomimetics-05-00046-f014:**
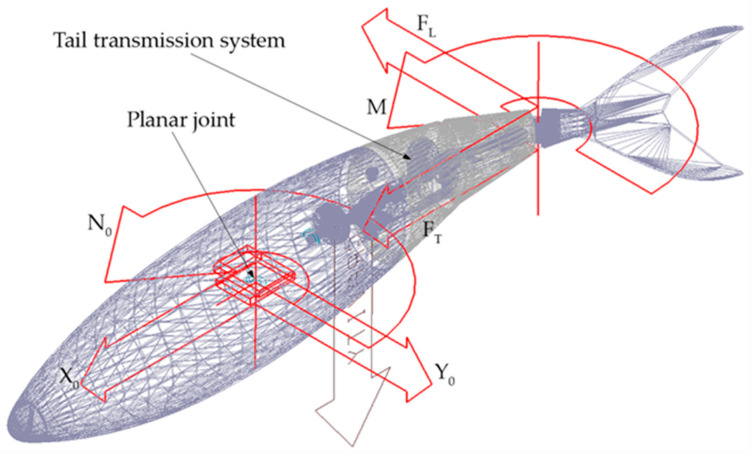
Multibody model of the carangiform swimming robot imported in Adams/View by MSC software^®^.

**Figure 15 biomimetics-05-00046-f015:**
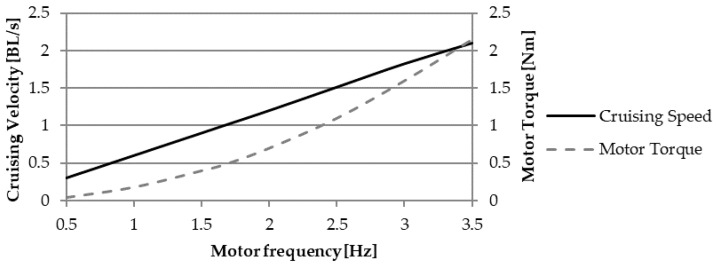
Average cruising speed and driving torque as a function of the motor frequency.

**Table 1 biomimetics-05-00046-t001:** Added mass and damping coefficient for a cylinder with a radius *R*, length *L*, and mass *m* [17].

$X_{\dot{u}}$	$Y_{\dot{v}}$	$N_{\dot{r}}$	$X_{u\left|u\right|}$	$Y_{v\left|v\right|}$
0.1 m	$\pi \rho R^{2}L$	$\pi \rho R^{2}L^{3}/12$	$\rho A_{f}c_{D,f}/2$	$\rho A_{l}c_{D,l}/2$
$N_{r\left|r\right|}$	$A_{f}$	$A_{l}$	$c_{D,f}$	$c_{D,l}$
$\rho A_{l}c_{D,l}L^{3}/16$	$\pi R^{2}$	2RL	0.5	[0.8–1.2]

**Table 2 biomimetics-05-00046-t002:** Oscillation amplitudes and phase shifts of the harmonics functions (9).

*A* _1_	*A* _2_	*A* _3_	Δ_1_	Δ_2_	Δ_3_
0.074	0.519	0.585	0	−0.49	−1.33

**Table 3 biomimetics-05-00046-t003:** Geometric parameters of the three *j*th cam joints; *h_j_* and *L_j_* are expressed in mm, *δ_j_* in radians.

*h* _1_	*L* _1_	*δ* _1_	*h* _2_	*L* _2_	*δ* _2_	*h* _3_	*L* _3_	*δ* _3_
4	54	0	11.8	26	−0.57	6.3	14	−2.35

**Table 4 biomimetics-05-00046-t004:** Reynolds numbers in the preliminary investigation.

**Reynolds Number**	**Propulsive Efficiency %**	**Thrust Coefficient**
1 × 10^4^	67.7	0.258
5 × 10^4^	68.3	0.261
1 × 10^5^	69.5	0.263
5 × 10^5^	70.1	0.265
1 × 10^6^	70.4	0.267

**Table 5 biomimetics-05-00046-t005:** Fish geometry, mass, and inertia properties, all expressed in SI units.

*c* _25%_	*c* _75%_	*B*	*m*	*I_xx_*	*I_yy_*	*I_zz_*
0.080	0.060	0.168	6.147	0.143	0.012	0.139

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
