# Peer review of "Design of a Carangiform Swimming Robot through a Multiphysics Simulation Environment"

_biomimetics, 2020, doi:10.3390/biomimetics5040046_

Round 1
Reviewer 1 Report
The paper is written well, the methods are explained clearly, and the results are interpreted carefully. The limitations (approximations) of the approach are mentioned as well. The results are interesting, and the manuscript fits well within the scope of “Biomimetics”.
Overall, solid work and I recommend publication in Biomimetics. One concrete remark: in the very last paragraph, the authors mention follow-up work. Rather than using phrases like “has been drawn already”; “in the following weeks”, I would phrase this more neutrally, e.g. “Future research will focus on …”.
Another remark, which may or may not be taken into account in the paper: from the robotics point-of-view, this work follows a conventional approach where stiff components are connected through (complex) hinges driven by motors and transmission systems. I’m sure the authors are aware of the relatively new development of “soft robotics” in which flexible, sometimes responsive, materials are used allowing for more “natural” complex motions by exploiting the deformability of these materials (e.g. G.M. Whitesides, Angew. Chem. Int. Ed. 2018, 57, 2 – 18). How does the authors relate their work to this development? Do they see an opportunity for the soft robotics approach to even better approximate the swimming gaits shown in the manuscript?
Author Response
Overall, solid work and I recommend publication in Biomimetics. One concrete remark: in the very last paragraph, the authors mention follow-up work. Rather than using phrases like “has been drawn already”; “in the following weeks”, I would phrase this more neutrally, e.g. “Future research will focus on …”.
We thank the reviewer for the suggestion. The part regarding future research has been corrected as suggested. We also added the following sentence:
Moreover, future works will be also focused on the design and simulation of a transmission system capable to drive the pectoral fins of the proposed prototype. The devised solution will allow the robot to change its course according to its mission.
Another remark, which may or may not be taken into account in the paper: from the robotics point-of-view, this work follows a conventional approach where stiff components are connected through (complex) hinges driven by motors and transmission systems. I’m sure the authors are aware of the relatively new development of “soft robotics” in which flexible, sometimes responsive, materials are used allowing for more “natural” complex motions by exploiting the deformability of these materials (e.g. G.M. Whitesides, Angew. Chem. Int. Ed. 2018, 57, 2 – 18). How does the authors relate their work to this development? Do they see an opportunity for the soft robotics approach to even better approximate the swimming gaits shown in the manuscript?
We agree with the Reviewer. The soft robotics design method and the suggested reference has been cited in the Introduction for the sake of completeness. Regarding the possibility to improve our design with a compliant tail, we are well aware that a more natural motion would be achieved by soft robotics solutions. However, we decided to complete the development of the spatial-cam joint transmission system before moving to a more complex design.
Reviewer 2 Report
Please see attached comments.

Author Response
We thank the reviewer for the suggested corrections.
Figure 2, 3 and 4 have been updated with the missing labels. We also cited equations (9) and (10) in the caption of Figure 12. Finally, references [1] and [6] have been added to the caption of Figure 6.
References doi:10.3390/app8020180 and doi:10.1177/1729881416669483 have been added to the paper as suggested.
Typos and clarity improvements have been corrected.
It’s mentioned that the thunniform swimming mode is more efficient that that predicted for the prosed robotic swimmer, but it also suggested to briefly discuss the predicted maneuverability difference between robotic thunniform swimmers and the carangiform swimmer presented.
We thank the Reviewer for the suggestion. The following sentence has been added to the final Discussion:
As a matter of fact, the design evolved by thunniform swimmers in terms of body shape and mass distribution is optimized for high-speed, efficient swimming while is not well-suited to other actions such as turning maneuvers and rapid accelerations. On the contrary, carangiform swimmers possess slender and flexible bodies which allows them to perform turnabout maneuvers characterized by a normalized radius of curvature lower than one body length [1].